# Prevalence of undiagnosed hypertension and risk assessment using a validated survey in community-based screening in Amman, Jordan

Anas Khaleel[ID][1], Malak Al-Quaiti[1], Sara Istaitiya[1], Abhijit V. Kshirsagar[2], Heejung Bang[3¤a*]

**1** Department of Pharmacology and Biomedical Sciences, Faculty of Pharmacy, University of Petra, Amman, Jordan, **2** Division of Nephrology and Hypertension, School of Medicine, University of North Carolina-Chapel Hill, Chapel Hill, North Carolina, United States of America, **3** Division of Biostatistics, Department of Public Health Sciences, University of California-Davis, Davis, California, United States of America

¤a Current Address: Division of Biostatistics, Department of Public Health Sciences, University of California-Davis, Davis, California, United States of America
* hbang@health.ucdavis.edu

## Abstract

### Background and objectives

The rising incidence of hypertension in Jordan has contributed to high rates of associated morbidity/mortality and increased societal costs. Hypertension affects approximately 30% of Jordan's adult population. This study assessed the prevalence of undiagnosed hypertension through blood pressure measurement and evaluated future hypertension risk using a validated survey in a community setting. The survey instrument incorporates demographic and clinical factors to predict the likelihood of hypertension development within a nine-year timeframe.

### Methods

We conducted a cross-sectional local community screening in Amman, Jordan (Sept 2022-Sept 2023). Trained university pharmacy students had measured blood pressure and distributed a validated 9-risk predictors questionnaire to adults without diagnosed hypertension during a single visit. Participants were categorized using American Heart Association criteria.

### Results

Among 932 adult participants of the study, approximately 57% were female, 34% were overweight, 16% were obese, and 5.3% were extremely obese. Risk factors included current smoking (38%), diabetes mellitus comorbidity (8%), family history of hypertension (61%), and insufficient exercise (62%). Systolic readings were classified as elevated (45%), stage 1 (15%), or stage 2 (5.5%). Diastolic readings had stage 1 (42%) or stage 2 (7.6%). Participants with risk scores ≥17 (n = 39) were those who

**Data availability statement:** All relevant data are within the manuscript and its Supporting information files.

**Funding:** H.B. was partly supported by the National Institutes of Health through grants UL1 TR001860. The content is solely the responsibility of the authors and does not represent the official views of the National Institutes of Health. The funders had no role in study design, data collection and analysis, decision to publish, or preparation of the manuscript. There was no additional external funding received for this study.

**Competing interests:** The authors have declared that no competing interests exist.

had current hypertension (BP ≥ 140/90 mmHg). Those were classified as high-risk based on established cutoffs.

## Conclusion

Our community screening revealed a high prevalence of undiagnosed hypertension (20.6%) and identified a substantial proportion at high risk for future hypertension. The risk assessment tool showed good screening ability to reveal the participants' hypertension status, supporting its utility for risk stratification in resource-limited settings. Moreover, our validated risk tool effectively classified between those with and without current hypertension and identified individuals needing intensive preventive interventions. This dual approach of BP measurement and risk assessment may guide targeted screening intervals and preventive interventions in Jordan and similar settings.

## Introduction

High blood pressure (BP) or hypertension, defined by the World Health Organization as a BP reading ≥140/90 mmHg, is a global health problem. It is a highly prevalent, chronic age-related disorder often contributing to cardiovascular and/or kidney complications [1–5]. The incidence of hypertension in Jordan has increased steadily in recent years–20% in 2012 to 30% by 2020 and expected to rise to 46% in 2030 [6–8].

Undiagnosed hypertension represents a hidden national health burden. Alhawari et al. found that nearly 38% of Jordanians had undiagnosed stage 1 hypertension, and 30% had undiagnosed stage 2 hypertension, according to the American College of Cardiology (ACC)/American Heart Association (AHA) guidelines [9]. There are many risk factors for hypertension beyond well-known factors of age and family history [10]. Being overweight and obese increases the risk of hypertension and cardiovascular disease (CVD). Lack of exercise or physical activity can cause weight gain, increasing hypertension risk [11–14]. Smoking also can cause an immediate increase in BP and is considered a contemporary risk factor for developing hypertension [15–18]. High salt intake from food causes the body to retain fluids, increasing BP and predisposing a person to develop hypertension [19–21]. High levels of stress and stress-related habits can also lead to a temporary increase in BP and may contribute to risk factors causing hypertension [22–25]. Certain chronic diseases or comorbidities, such as diabetes mellitus (DM) and chronic kidney disease (CKD), can also lead to high BP [26,27]. It has been reported that DM was significantly associated with hypertension [8].

Internationally, risk-scoring systems have been developed to identify high-risk individuals via estimating the probability of developing hypertension (or other critical medical conditions). These tools aim to increase patient awareness and health education, which could help reduce the prevalence of hypertension and control comorbidities and incidence/recurrence of CVD [28,29]. In CVD, DM, and CKD, risk

assessment tools are widely used, and some have been cross-validated in Western and Eastern populations [30,31]. In contrast, self-assessment of risk in practice is not commonly utilized for hypertension despite availability [32], perhaps because BP can be directly measured. Additionally, the general public is not well aware of the future risk of events associated with hypertension.

Awareness of hypertension varies widely in the Middle East; Oman (24%), Morocco (22%), and Jordan (14% to 82%) and likely depends on the age of participants in different awareness studies [33]. Reports have confirmed that individuals from lower socioeconomic backgrounds are more likely to have undiagnosed hypertension [34,35].

Thus, there is a need to screen for hypertension risk in Jordan. Risk assessment instruments can be used to identify at-risk individuals, and to implement preventive strategies, including periodic blood pressure monitoring or altering modifiable risk factors/lifestyles to avoid or delay hypertension onset [32]. Low-cost treatments, counseling, and preventive approaches are available throughout Jordan. Preventive programs showed that post-campaign response can improve lifestyle and dietary habits and awareness among Jordanians [36].

The current study aimed to identify the prevalence of undiagnosed hypertension and assess future hypertension risk using a validated risk-scoring tool in the general public in Amman, Jordan.

## Methods

### Study design, setting, and participants

Pharmacy students from the University of Petra volunteered to participate as data collectors. They administered questionnaires to family members, friends, and neighbors who met the inclusion criteria (adults without known hypertension). During the same visit, students measured participants' blood pressure and assisted with questionnaire completion. This convenience sampling approach limits generalizability beyond the study population. The structured questionnaire was used to gather data on sociodemographic, clinical, and behavioral/lifestyle factors. A total of 1000 questionnaires were distributed to University of Petra Pharmacy College students and collected from 07/09/2022 to 07/09/2023. Supporting file 1 contains the study IRB document. Students participated in this study and also distributed questionnaires randomly to their family members and friends/neighbors and were instructed to invite persons without *known* hypertension (i.e., taking hypertension medication or having been diagnosed by a healthcare provider).

Questionnaires were devised to assess the future risk of developing hypertension via questions included in Kshirsagar et al.'s risk scoring [32] and additional background and potential risk factors. Supporting file 2 contains the original questionnaire.

The survey was translated into Arabic and tested by expert faculty members and students for comprehension and context. Five faculty members read and revised questions and verified their validity and ease of understanding without response bias. Participants (~30 individuals) in a pilot testing also provided feedback regarding the clarity of questions. Written informed consent was provided by all participants who were reminded that participation was entirely voluntary and that they could opt-out at any time and for any reason. No identifying information was collected; all questionnaires were anonymous. This study design was approved by the Institutional Review Board of the University of Petra (decision number: 20220443).

Pharmacy students helped to measure and record resting BP using Omron™ BP measuring electronic machine (Omron Healthcare, Osaka, Japan); model Omron HEM-907XL IntelliSense Professional and to assist participants in completing questionnaires. Students received brief training from two college pharmacy professors on correctly measuring BP and individuals being seated. Moreover, participants were instructed to abstain from smoking and caffeine intake for at least 30 minutes before measuring. BP was measured once; if the reading was elevated (SBP ≥ 120 or DBP ≥ 80 mmHg), a second measurement was taken after 1–2 minutes of rest, and the two readings were averaged. Rarely, three measurements were taken and averaged. This pragmatic approach balanced resource constraints with measurement reliability but introduced potential variability due to non-standardized intervals between readings. Limitations in a community-based

study (long-term follow-up) and rational multiple readings are standard for BP diagnosis. Data were entered into Microsoft Excel, cleaned, and analyzed.

This electronic machine model has been validated for clinical use according to international protocols and is recommended by the European Society of Hypertension for office BP measurements.

Standardized multiple readings at fixed intervals (e.g., 3 readings at 1-minute intervals) are ideal for BP diagnosis. Moreover, data collection occurred in a single visit where participants completed the questionnaire with assistance from trained pharmacy students who also measured their BP during the same session.

**Risk assessment tool.** The risk assessment tool we utilized was adapted from one originally developed for the US adult population using large prospective cohorts and combined risk factors, which often co-exist and cumulatively affect the risk of hypertension; it provides individual item and total integer scores and can be converted to risk probabilities in a user-friendly format [32]. Our questionnaire included 15 questions, including demographics (six questions) and nine variables/predictors to assess the general public's risk (or probability) of developing hypertension up to 9 years hence.

Risk factors with corresponding points are provided below:

1. Age (years): ≤54 y = 0 points; 55–64 y = 2 points; 65–74 y = 3 points; and ≥75 y = 4 points.

2. Sex (female): yes = 1, no = 0 points.

3. Smoking status: yes = 1, no = 0 points.

4. Exercise: yes = 0, no = 1 points.

5. Family history of hypertension: yes = 1, no = 0 points.

6. Body mass index: <25 (normal weight)=0; 25–29 (overweight)=1; 30–39 (obese)=2; and ≥40 (extremely obese)=3 points.

7. Diabetes: yes = 1, no = 0 points.

8. BP readings:

    a. Systolic BP (SBP) in mmHg

        i. SBP <110 = 0; 110–114 = 2; 115–119 = 3; 120–124 = 4; 125–129 = 6; 130–134 = 8; 135–139 = 14; and >140 = 14 points.

    b. Diastolic BP (DBP) in mmHg, depending on age

        i. For age <55 y; DBP <70 = 0, 70–79 = 2, ≥80 = 3 points.

        ii. For age in 55–64 y; DBP 70–79 = −1, ≥80 = −1 points.

        iii. For age in 65–74 y; DBP 70–79 = −2, ≥80 = −3 points.

        iv. For age ≥75 y; DBP 70–79 = −1, ≥80 = −2 points.

A total risk score was calculated as the sum of all points, and this value was translated to the risk of hypertension over the next 3, 6, and 9 years using the published risk chart.

Age, level of systolic or diastolic BP, smoking, family history of hypertension, DM, body mass index, sex, and exercise were associated with the development of hypertension [32]. We used the original risk score, without adding other potential risk factors where some are hard to measure in the community settings. A score of ≥17 points was considered a high risk of developing hypertension according to the cutoff value proposed by Kshirsagar et al.'s validated risk assessment tool, developed and applied previously in the US population and currently to the Jordanian population [32]. The minimum obtainable score is 0 points, and the maximum is 30 points.

**Statistical analyses.** Statistical analyses were performed using the Statistical Package for Social Science version 25 (SPSS Inc., Chicago, IL, USA). We used descriptive statistics to summarize data: continuous variables were used to summarize age and hypertension risk via mean or median and standard deviation or range, and categorical variables were presented as frequencies and percentages. Missing data and incomplete questionnaires were excluded from statistical analysis. Risk scores were obtained to stratify individuals for the risk of hypertension incidence.

## Results

### Characteristics of participants

A total of 932 adults participated in this study. Table 1 presents participant characteristics. Study participant selection and analysis are shown in Fig 1. A total of 1000 questionnaires were distributed, and 932 individuals (~93% response rate) participated and were included in the analyses; 68 were excluded due to incomplete/missing data. Participants ranged from 18–89 years old, with a median age of 40. Among 932 participants, 402 (43%) were male, and 530 (57%) were female. Current smokers were 38%. Consistent with exclusion criteria, no participant was taking BP medication or hypertension-related treatment.

Among the 932 participants without diagnosed hypertension, SBP readings showed that 34% were normal, 45% were elevated, 15% were stage 1, and 5.5% were stage 2; for DBP, 50% were normal or elevated, 42% were stage 1, and 7.6% were stage 2, according to the AHA guidelines (Table 2). Prehypertension (elevated BP) per AHA refers to BP higher than usual; readings can range from 120 to 129 for SBP with normal DBP [37]. As such, some participants newly learned about their hypertension status as a result of this risk assessment survey.

**Risk scores that predict hypertension at 3, 6, and 9 years.** Total scores were calculated as the sum of points obtained from the risk assessment questionnaires. These scores were used to estimate the risk probability for the next 3, 6, and 9 years, following risk tabulation published by Kshirsagar et al. [32]. As detailed in S1 Table. Fig 2 summarize risk scores and future risk of developing hypertension. Participants who scored 11 points have a 9-year risk of >50%, and participants who scored 15 points have a 6-year risk of >50%. Additionally, participants who scored 22 points have three 3-year risk >50%. With the cutoff point of 17 suggested from the original model/study for the US population, the 6-year risk is almost 50%, and the 9-year risk is 75%.

We found that 51 participants (5.5%) were classified as stage 2 hypertension according to the AHA criteria, 141 participants (15.1%) were classified as stage 1 hypertension, 423 participants (45.4%) had elevated BP, and 314 participants (33.7%) had normal BP based on both readings SBP and DBP (SBP<120 mmHg and DBP<80 mmHg). When we analyzed the DBP readings separately, we observed that 71 participants (7.6%) had stage 2 hypertension, 391 participants (42.0%) had stage 1 hypertension, and 470 participants (50.4%) had normal readings. Taking SBP and DBP together, we found that three out of 932 participants constitute 0.3% recognized to have a hypertensive crisis (all 3 had SBP>180 mmHg and 0 had DBP>120 mmHg); those participants were told that their reading was very high so they should seek medical attention as soon as possible. Also, 423 individuals (45.4%) with elevated BP based on SBP (in 120–129 mmHg) were identified in the current study.

Additionally, among the 39 participants with risk scores ≥17, the following characteristics were observed:

- Mean age: 58 years (SD ± 9)

- Female: 38%

- Current smokers: 29%

- Family history of hypertension: 72%

- BMI ≥ 30: 54%

- DM: 31%

- No regular exercise: 67%

**Table 1. Sociodemographic and CLINICAL CHARACTERISTICS OF STUDY PARTICIPANTS (N = 932).**

| Characteristics | n (%) |
|---|---|
| **Age, years** | median: 40 (range: 18–89) |
| **Sex** | |
| Male | 402 (43.1%) |
| Female | 530 (56.9%) |
| **Education level** | |
| Less than High school | 216 (23.2%) |
| High school diploma | 84 (9.0%) |
| Bachelor | 551 (59.1%) |
| Master's degree | 49 (5.3%) |
| Doctorate | 32 (3.4%) |
| **Medical degree or major*** | |
| Yes | 186 (20.0%) |
| No | 746 (80.0%) |
| **Medical insurance** | |
| Yes | 490 (52.6%) |
| No | 442 (47.4%) |
| **Marital/living status** | |
| Married | 519 (55.7%) |
| Single | 360 (38.6%) |
| Divorced/widow | 53 (5.7%) |
| **Monthly Salary Jordanian Dinar (JD), 1 JD = 0.71 USD (2023 rate)** | |
| Less than 250 JDs | 50 (5.4%) |
| 251–500 JDs | 90 (9.7%) |
| 501–750 JDs | 64 (6.9%) |
| 751–1000 JDs | 49 (5.3%) |
| More than 1000 JDs | 82 (8.8%) |
| Refused to answer | 597 (64.1%) |
| **Body Mass Index (kg/m²)** | |
| Underweight | 19 (2.0%) |
| Normal | 397 (42.6%) |
| Overweight | 317 (34.0%) |
| Obese | 150 (16.1%) |
| Extremely obese | 49 (5.3%) |
| **High salt intake > 5g per day (about 3/4 of a teaspoon)** | |
| Yes | 536 (57.5%) |
| No | 396 (42.5%) |
| **Smoking status** | |
| Current smoker | 353 (37.9%) |
| Former smoker | 26 (2.8%) |
| Never | 553 (59.3%) |
| **Diabetes mellitus** | |
| Yes | 74 (7.9%) |
| No | 858 (92.1%) |

*(Continued)*

**Table 1.** (Continued)

| Characteristics | n (%) |
|---|---|
| **Family history of hypertension**\*\* | |
| Yes | 564 (60.5%) |
| No | 368 (39.5%) |
| **Exercise status**\*\*\* | |
| Yes | 352 (37.8%) |
| No | 580 (62.2%) |

\*Medical major: MD/nurse/pharmacy, etc.

\*\*Family history of hypertension: reporting one or two parents who have been diagnosed with hypertension.

\*\*\*30–60 minutes of physical activity at least three days per week (approximately 140 minutes of moderate exercise, according to the American College of Cardiology).

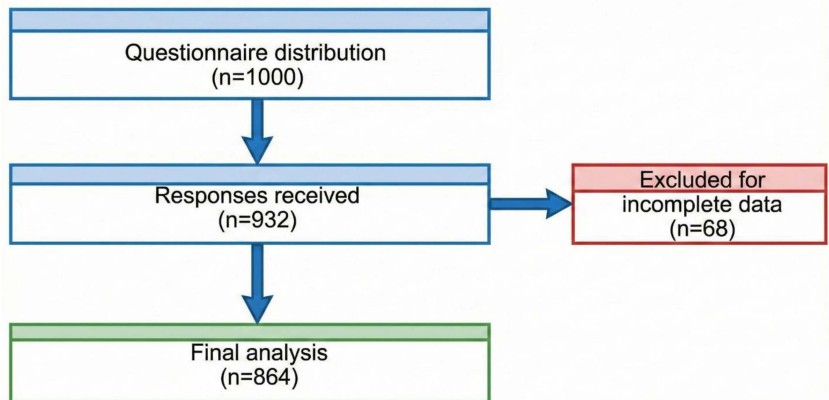

**Fig 1. The chart outlines the selection process for participants, starting with the initial distribution of questionnaires through the final analysis phase.** One thousand questionnaires were sent to adults who had never been diagnosed with hypertension. A total of 932 subjects were included in the final analysis.

**Table 2. Blood pressure measurements and distribution (N = 932).**

| Systolic Blood Pressure (in mmHg) | |
|---|---|
| • Normal <120 | 314 (33.7%) |
| • Elevated 120–129 | 423 (45.4%) |
| • Stage 1 130–139 | 141 (15.1%) |
| • Stage 2 ≥ 140<br>• Hypertensive crisis >180 | 51 (5.5%)<br>3 (0.3%) |
| **Diastolic Blood Pressure** (in mmHg) | |
| • Normal & Elevated <80 | 470 (50.4%) |
| • Stage 1 80–89 | 391 (42.0%) |
| • Stage 2 ≥ 90 | 71 (7.6%) |
| • Hypertensive crisis >120 | 0 (0%) |

Per the American Heart Association, systolic blood pressure (SBP) less than 120, and diastolic blood pressure (DBP) less than 80 is considered normal. Elevated BP is SBP between 120 and 129, even with normal DBP. High BP (hypertension) is classified by stage: Stage 1: SBP is 130–139 or DBP 80–89, and Stage 2: SBP is 140 or higher, or DBP is 90 or above. In clinical practice, clinicians usually use combined readings (when assessing patients' BP) and categorize patients based on the higher reading level, either SBP or DBP. Multiple measurements in different days (occasions) are required to confirm the clinical diagnosis.

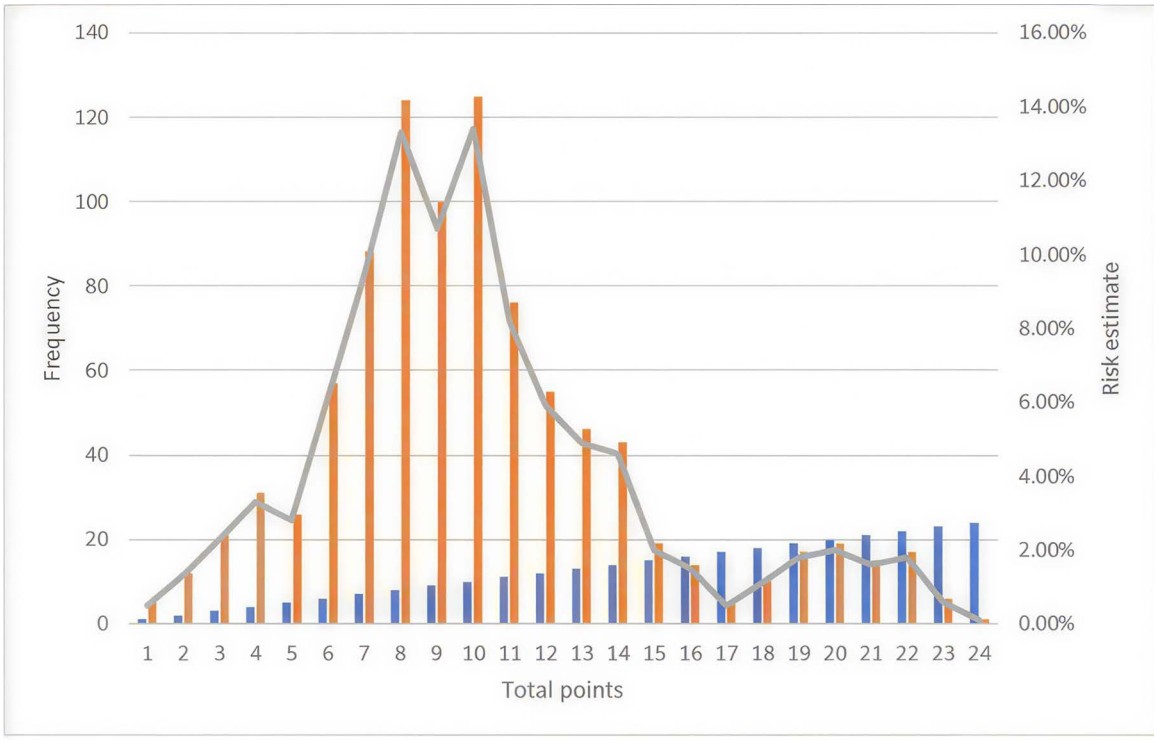

**Fig 2. Total scores and risk estimates of the 932 participants.** Orange bars represent the frequency (number of participants in the risk category). Blue bars represent the total risk points scored. The grey line represents the risk percentage among participants.

This approach could optimize resource allocation in resource-limited settings while ensuring timely detection of hypertension among those at highest risk.

To examine the relationship between high-risk scores and current hypertension status, we conducted a 2x2 contingency table analysis.

Among participants with risk scores ≥17 (n = 39), those who had current hypertension (BP ≥ 140/90 mmHg) and/or the high-risk and lower-risk participants are summarized in Table 3.

The risk score demonstrated screening utility for identifying current hypertension, with an odds ratio (OR) of 19.3 (95% CI: 9.2–40.4; p < 0.001), sensitivity of 62.8%, and specificity of 96.2%. These results enhance the tool's utility for risk stratification rather than definitive diagnosis.

**Table 3. Contingency table for high risk score >17 points.**

| | Hypertension Present (BP ≥ 140/90) | Hypertension Absent (BP < 140/90) | Total |
|---|---|---|---|
| **High Risk (>17)** | 23 | 16 | **39** |
| **Low/Mod Risk (≤17)** | 31 | 862 | **893** |
| **Total** | **54** | **878** | **932** |

OR: 19.3 (95% CI: 9.2–40.4, p < 0.001), Sensitivity: 62.8%, Specificity: 96.2%, PPV: 64.8%, NPV: 95.8%.

## Discussion

In this community-based study for hypertension screening and risk assessment, we employed a risk-scoring tool for a convenience sample of 932 Jordanian adults. We utilized a risk assessment tool adapted from a validated instrument to predict the likelihood of hypertension development within a nine-year period. This tool incorporates a comprehensive set of risk factors, including demographics and lifestyle variables.

We found that nearly 70% of our study participants (a cumulative proportion on elevated, stage 1 and stage 2 readings on SBP is 45.4 + 15.1 + 5.5%, respectively; plus, 5% separately high DBP reading) could have met the condition of prehypertension or hypertension, according to the AHA BP reading criteria [37,38]. According to Khader et al., the prevalence of hypertension in the general Jordanian population is estimated as one in every three adults; specifically, 29% of adult females and 34% of adult males were officially diagnosed and taking antihypertensive medications [8]. Nevertheless, this is similar to the prevalence of hypertension in the wider Arab region, reported 29% [8], comparable to the US prevalence of 32%, but lower than the rate in European countries, which is 39% (not adjusting calendar years) [39].

The detection of new cases (some urgent) in this study demonstrates the importance of hypertension screening to increase awareness and detection of hypertension and associated risk factors, and to play a critical role in communication and shared decision-making between clinicians and patients. The high rates of modifiable risk factors, namely physical inactivity (62%), smoking (38%), and obesity (21%), show that there are high opportunities for targeted interventions. The existing tobacco control programs in Jordan will be of special benefit to this group. Notably, 16/39 (41%) high-risk scorers did not have current diagnosed hypertension, representing a significant prevention opportunity. These individuals require intensive lifestyle counseling and frequent BP monitoring.

Our study also demonstrates a potential role for community pharmacies, pharmacy interns and students to help identify/screen for undiagnosed hypertension and assess/teach future risk. Community pharmacies−together with pharmacists or pharmacy assistants−can serve as affordable, friendly and accessible locations, especially for low socioeconomic, low health literacy and busy citizens of Jordan. Simple pencil and paper risk assessment tools and BP machines may be easily and inexpensively installed in waiting rooms in clinic and community health center as well as at pharmacies and health fairs/events.

Our report showed that ~45% (423/932) of participants likely have prehypertension based on SBP readings. Those individuals were to be referred to primary care or special clinics for accurate diagnosis or for follow-up and monitoring of BP at home and with clinician (and related comorbidities) to prevent developing chronic hypertension or for general health management according to current medical guidelines [40]. Prehypertension is a significant public health concern in Jordan, with a considerable portion of the population falling within this category. Many individuals with prehypertension may be unaware of their condition, leading to delayed diagnosis and intervention. Prehypertension is treated with lifestyle changes, and sometimes medication or nonpharmacologic therapy according to AHA/ACC.

Preventive strategies may be inferred from the scoring system to help patients recognize hypertension−possibly with a healthcare worker's help−and associated risk factors. A previous report showed that females and higher education levels were positively associated with diets specific to lowering or controlling BP [41]. Another study highlighted the importance of patient self-motivation in managing hypertension. The authors suggest that clinicians and other healthcare providers should recognize that patients may respond differently regarding treatment compliance [42]. *Personalized* plans to prevent or treat hypertension and related chronic diseases such as DM and CKD jointly can involve a long-term commitment both from patients/family as well as clinicians to lifestyle modification and pharmacologic therapy. One group that likely needs personalized plans with frequent blood pressure monitoring are those that score ≥ 17, at least every 3 months to determine if therapy is needed. The good news is that many medications are available, safe, and widely accessible at low cost in Jordan. Based on our findings, we propose a risk-stratified screening approach: High risk (score ≥17), moderate risk (score 11–16), and finally low risk (score <11).

The prevalence of hypertension in Arab countries is approximately 30% [7,39]. Health education, therapeutic lifestyle and environment changes, counseling, and behavior modifications can improve the awareness of cardiac-related conditions (and important roles of BP), attitudes, and responsibility for daily health management at home, nutritional practices, and interpersonal relationships [36]. Various strategies to decrease CVD, DM, and hypertension risk have been published [43,44]. including prevention and treatment of obesity, physical activity (such as aerobic exercise, walking, or swimming for 30–60 minutes at least three days/week), reducing dietary salt, total fat, and cholesterol, limiting caffeine, alcohol, red meat, sweets and sugary beverages, avoiding smoking, and managing stress [45]. Notably, the same risk factors contribute to diseases such as CVD, DM, and CKD. Inter-relationships among these medical conditions are well established and risk predictions for them have been actively researched. Since hypertension is a *common* risk factor in all of these conditions, BP measurement and education could be a *first* step to prediction, risk assessment and prevention and management of CVD, DM and CKD. The validated risk tool effectively stratified individuals by hypertension risk and identified those with current elevated BP. While not a diagnostic tool, it shows promise for community-based screening and triage in resource-limited settings

## Limitations

Incomplete surveys were a limitation in this study; Approximately 1000 surveys were distributed, and 932 participants answered all of the questions. Because questionnaires were anonymous, we were unable to contact participants for follow-up; we wish to conduct a longitudinal or prospective study in the future to ascertain predictive accuracy. Students received short training on BP measurement technique but no formal proficiency assessment was conducted. This may have introduced inter-observer variability in measurements. Not all BP elevators were considered in this study, including stress, caffeine, and alcohol intake, which are difficult to measure accurately. Exclusion of stress as a factor for elevating BP, is also explaining its methodological challenges. Previously, there were no unique determinants or risk assessments for the Jordanian population to reflect on, so we used tools from abroad. Next, a single or multiple BP measurement collected with the BP machine Omron™ from a majority of participants at a single time point for this community study could be insufficient to detect hypertension accurately (office use in this setting). This approach of measuring was a compromise to balance resource constraints and measurement reliability in a large-scale community-based study. Lastly, a convenience sample may not represent a target population, such as the wider university family, Amman, or Jordanian population.

Yet, the wide age range and a relatively large sample size of over 900 participants representing diverse sociodemographic groups can be a strength. Utilizing pharmacy students to assist with BP measurement and survey completion adds to the quality and reliability of data; if participant's self-assessment was utilized alone, measurement error/misclassification would be more likely but wider implementation is possible. These study features may inform future design and planning of community screening programs and toward an optimal use of patient-centered risk assessment tools. To our knowledge, this is the first study of its kind in Arab Middle Eastern country, Jordan, and may provide a model example for real-world screening, risk assessment, referral for treatment/clinician, and research. Jordan's National Strategy for Non-Communicable Diseases recommends opportunistic BP screening in primary care settings. However, systematic community-based screening programs remain limited, highlighting the need for our approach. The lack of a standardized protocol for multiple BP measurements at fixed intervals may have introduced measurement variability, though we attempted to minimize this by taking additional readings for abnormal results.

Additional limitations include potential recall bias in self-reported variables such as exercise habits, which may lead to underestimation of these risk factors. Furthermore, the variability in BP measurement methods, including the lack of standardized timing between readings and potential differences in cuff positioning or participant preparation across multiple student measurers, may have introduced measurement bias. Although students received training, inter-observer variability cannot be excluded."

## Conclusion

Hypertension is a vastly important but poorly controlled public health problem in Amman, Jordan. Almost 70% of study participants met the AHA criteria for a diagnosis of prehypertension or hypertension but they were likely unaware of their (pre)hypertension status. Policymakers and healthcare practitioners in Jordan must discuss an evidence-based practice effort to prevent and test for hypertension in a larger scale and associated health services issues. We used a validated tool to predict/estimate hypertension risk and collect subjects' current BP along with demographic and clinical risk factors. Our study demonstrated the feasibility of using a risk-scoring instrument in a community setting and generated some valuable knowledge for the purposes of health education, screening, and prevention. Our team hopes to develop targeted screenings for hypertension and related chronic diseases and to evaluate the risk-benefit and cost-effectiveness in Jordan, which could inform similar efforts in resource-limited settings worldwide.

Transparent, evidence-based, validated, and easily accessible tools could be helpful to facilitate patient-provider communication. Used as a low-cost, first-step approach, the self-administered (pre)screening tool can raise public awareness about hypertension, improve BP control, delay or prevent hypertension, and potentially reduce high-cost cardiovascular and renal complications in Jordan and beyond.

## Supporting information

**S1 Table. 3-, 6-, and 9-year Risk of Incident Hypertension by Baseline Factors and Total Risk Score.**
(PDF)

**S2 Table. Raw data variables.**
(XLSX)

**S1 Questionnaire. Hypertension screening survey.**
(PDF)

## Author contributions

**Conceptualization:** Anas Khaleel.

**Formal analysis:** Malak Al-Quaiti, Sara Istaitiya.

**Writing – review & editing:** Anas Khaleel, Abhijit V. Kshirsagar, Heejung Bang.

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
