## [Decision Letter · Decision Letter 0]

8 Jul 2025

Dear Dr. Khaleel,

Thank you for submitting your manuscript to PLOS ONE. After careful consideration, we feel that it has merit but does not fully meet PLOS ONE’s publication criteria as it currently stands. Therefore, we invite you to submit a revised version of the manuscript that addresses the points raised during the review process.

We look forward to receiving your revised manuscript.

Kind regards,

Zypher Jude G. Regencia, Ph.D.

Academic Editor

PLOS ONE

Journal Requirements:

“H.B. was partly supported by the National Institutes of Health

through grants UL1 TR001860. The content is solely the responsibility of the authors and does

not represent the official views of the National Institutes of Health”

5. Thank you for stating in your Funding Statement:

“H.B. was partly supported by the National Institutes of Health

through grants UL1 TR001860. The content is solely the responsibility of the authors and does

not represent the official views of the National Institutes of Health”

6. In the online submission form, you indicated that “The datasets used and analyzed for this study are available from the first author upon reasonable request.”

Reviewers' comments:

Reviewer's Responses to Questions

**Comments to the Author**

1. Is the manuscript technically sound, and do the data support the conclusions?

Reviewer #1: Yes

Reviewer #2: Yes

2. Has the statistical analysis been performed appropriately and rigorously?

Reviewer #1: No

Reviewer #2: Yes

3. Have the authors made all data underlying the findings in their manuscript fully available?

Reviewer #1: Yes

Reviewer #2: Yes

4. Is the manuscript presented in an intelligible fashion and written in standard English?

Reviewer #1: Yes

Reviewer #2: Yes

Reviewer #1: Overall, good paper but major improvements can be made to improve overall output

However, there are a few clarifications:

1. the objective states "identify undiagnosed cases of hypertension in the general public

using a risk-scoring tool for elevated BP and hypertension"

- it wasn't clear in the methods how the use of risk scoring tool identified undiagnosed cases of hypertension

- the methods essentially said that the study got the risk score and got the BP (and not used the risk score to identify undiagnosed hypertension as the objective and title imply

that being said may be the title and objective should reflect that was an identification of the prevalence of hypertension and estimating risk using the risk-scoring tool.

2. There was no standardization of the BP app used - pls indicate the model and why this was used - cite references that it is validated for office use

3. There was no standardization on the times BP was taken. Please indicate reason why this was not protocolized

4. Results separated systolic and diastolic hypertension. Is there a reason for this? Why not use the standard definitions only for hypertension as the summary result?

5. Suggest to do analytical statistics correlating HIGH risk score (>17) with incident hypertension at present (BP >140/90). This is a simple 2x2 table

6. Suggest to include analytics of the profile of those with HIGH score >17, and use that as jumping point for meaningful discussion on the timing of screening using the questionnaire.

7. Improve discussion to include suggested analytical analysis

8. Suggest to include recall bias in the limitations of the study (there were questions about salt intake and exercise) and bias of the methods of BP measurement

9. In conclusion gramatical error: switch "unlikely aware" to "likely unaware"

10. Improve abstract based on ALL comments above

Reviewer #2: Here are a few comments for the attention of the Authors:

After reading this part of the methods “Students participated in this study and also distributed questionnaires randomly to their family members and friends/neighbors and were instructed to invite persons without known hypertension” I am still confused as to what kind of recruitment strategy was used to select the students (I see serious selection bias)

Its is not very clear how the study process went. Questionnaires were administered first? Then they came to where to get BP measured? Please clarify these in the write up

The protocol followed in assessing the blood pressure doesn’t seem quite standard. Why are some people measured once, some twice and some thrice? Meanwhile, what is the meaning of abnormal reading? Please state the reference for the protocol used.

In the last sentence of the statistical analyses, it was stated that A score of ≥17 points was considered a high risk of developing hypertension according to the cutoff value proposed by Kshirsagar et al.’s validated risk assessment tool, developed and applied previously in the US population and currently to the Jordanian population.” This shouldn’t be a part of the statistical analyses. Rather it should be moved upwards and more detail provided. E.g. what is the minimum score obtainable?

No need repeating the entire results in the Table and the narration. E.g. male and female percentages (one is enough).

Results of those in Amman and other provinces is not relevant.

Why is there a citation at the last sentence of the results section?

Conclusion should be kept short. Most of the content there should go to discussion

**Do you want your identity to be public for this peer review?** For information about this choice, including consent withdrawal, please see our Privacy Policy

Reviewer #1: No

Reviewer #2: **Yes:** Jibril Mohammed

- - - - -

---

## [Author Response · Author response to Decision Letter 1]

10 Sep 2025

Point-by-Point Response to the Reviewers

Response to Reviewers

Manuscript ID: PONE-D-25-06643

Title: Use of a Risk Assessment Questionnaire to Identify Individuals with Hypertension in a Community-based Screening in Amman, Jordan

We thank the Editor and Reviewers for their time and constructive comments, which have helped us to improve the quality and clarity of our manuscript. Below we provide a detailed, point-by-point response. Reviewer comments are reproduced in bold, followed by our responses in regular font. All changes have been incorporated in the revised manuscript.

Response to Editor

• E1: Please ensure that your manuscript meets PLOS ONE's style requirements, including those for file naming. The PLOS ONE style templates can be found at https://journals.plos.org/plosone/s/file?id=wjVg/PLOSOne_formatting_sample_main_body.pdf and https://journals.plos.org/plosone/s/file?id=ba62/PLOSOne_formatting_sample_title_authors_affiliations.pdf

Response E1: All manuscript revised, style now according to journal requirement.

• E1: Please include a complete copy of PLOS’ questionnaire on inclusivity in global research in your revised manuscript. Our policy for research in this area aims to improve transparency in the reporting of research performed outside of researchers’ own country or community. The policy applies to researchers who have travelled to a different country to conduct research, research with Indigenous populations or their lands, and research on cultural artefacts. The questionnaire can also be requested at the journal’s discretion for any other submissions, even if these conditions are not met. Please find more information on the policy and a link to download a blank copy of the questionnaire here: https://journals.plos.org/plosone/s/best-practices-in-research-reporting. Please upload a completed version of your questionnaire as Supporting Information when you resubmit your manuscript.

Not relevant

• E1: We note that the grant information you provided in the ‘Funding Information’ and ‘Financial Disclosure’ sections do not match. When you resubmit, please ensure that you provide the correct grant numbers for the awards you received for your study in the ‘Funding Information’ section.

We are arranging that in resubmission, we will confirm the funding information.

• E1: Thank you for stating the following financial disclosure: “H.B. was partly supported by the National Institutes of Health through grants UL1 TR001860. The content is solely the responsibility of the authors and does not represent the official views of the National Institutes of Health” Please state what role the funders took in the study. If the funders had no role, please state: "The funders had no role in study design, data collection and analysis, decision to publish, or preparation of the manuscript." If this statement is not correct you must amend it as needed. Please include this amended Role of Funder statement in your cover letter; we will change the online submission form on your behalf.

All correct statements of funders and their roles are mentioned, in cover letter and manuscript and submission system.

• E1:Thank you for stating in your Funding Statement: “H.B. was partly supported by the National Institutes of Health through grants UL1 TR001860. The content is solely the responsibility of the authors and does not represent the official views of the National Institutes of Health” Please provide an amended statement that declares *all* the funding or sources of support (whether external or internal to your organization) received during this study, as detailed online in our guide for authors at http://journals.plos.org/plosone/s/submit-now. Please also include the statement “There was no additional external funding received for this study.” in your updated Funding Statement.

Included

• E1:Please include your amended Funding Statement within your cover letter. We will change the online submission form on your behalf.

Included

• E1:In the online submission form, you indicated that “The datasets used and analyzed for this study are available from the first author upon reasonable request.” All PLOS journals now require all data underlying the findings described in their manuscript to be freely available to other researchers, either 1. In a public repository, 2. Within the manuscript itself, or 3. Uploaded as supplementary information. This policy applies to all data except where public deposition would breach compliance with the protocol approved by your research ethics board. If your data cannot be made publicly available for ethical or legal reasons (e.g., public availability would compromise patient privacy), please explain your reasons on resubmission and your exemption request will be escalated for approval.

We chose: 3. Uploaded as supplementary information.

• E1:Please include captions for your Supporting Information files at the end of your manuscript, and update any in-text citations to match accordingly. Please see our Supporting Information guidelines for more information: http://journals.plos.org/plosone/s/supporting-information.

The manuscript has been entirely revised to meet the formatting requirements specified by the journal.

Response to Reviewers

Reviewer #1:

Overall, good paper but major improvements can be made to improve overall output

However, there are a few clarifications:

• R1: the objective states "identify undiagnosed cases of hypertension in the general public using a risk-scoring tool for elevated BP and hypertension" - it wasn't clear in the methods how the use of risk scoring tool identified undiagnosed cases of hypertension - the methods essentially said that the study got the risk score and got the BP (and not used the risk score to identify undiagnosed hypertension as the objective and title imply that being said may be the title and objective should reflect that was an identification of the prevalence of hypertension and estimating risk using the risk-scoring tool.

We appreciate this important clarification. You are correct that our original phrasing was misleading. We have revised both the title and objective to accurately reflect our study design:

Revised Title: "Prevalence of Undiagnosed Hypertension and Risk Assessment Using a Validated Questionnaire in Community-based Screening in Amman, Jordan"

Revised Objective: "The current study aimed to identify the prevalence of undiagnosed hypertension and assess future hypertension risk using a validated risk-scoring tool in the general public in Amman, Jordan."

We have also clarified in the abstract and methods that we conducted BP measurements to identify current undiagnosed hypertension AND separately used the risk-scoring tool to assess future hypertension risk among participants. This dual approach provides both immediate clinical value (identifying those who need treatment now) and preventive value (identifying those at high risk who need monitoring and lifestyle interventions).

Standardization of BP measurement device

• R1: There was no standardization of the BP app used - please indicate the model and why this was used - cite references that it is validated for office use.

Thank you for highlighting this important methodological detail. We have added the following information to the Methods section:

Pharmacy students helped to measure and record resting BP using the Omron™ BP measuring electronic machine (Model: LM Model, Approval No: IND/09/21/814, Omron Healthcare, Osaka, Japan). This model has been validated for clinical use according to international protocols and is recommended by the European Society of Hypertension for office BP measurements. Reference is ESH (European Society of Hypertension), also OMRON products are validated by the American Medical Association, see https://omronhealthcare.com/clinical-validation

• R1: There was no standardization on the times BP was taken. Please indicate reason why this was not protocolized.

We acknowledge this limitation and have expanded our discussion (page-14) and amended protocols in the Methods section (page-5):

"BP was generally measured once; if the result was abnormal (SBP ≥130 mmHg or DBP ≥80 mmHg), a second reading was taken after a 1-2 minute rest period, and then the two readings were averaged. Infrequently, three readings were taken and averaged when there was >10 mmHg difference between the first two readings. While standardized multiple readings at fixed intervals (e.g., 3 readings at 1-minute intervals) are ideal for BP diagnosis, this approach was not feasible (extra burden and inconvenient) in our community-based screening setting due to time constraints and the large number of participants. This pragmatic approach balances accuracy with feasibility in resource-limited community settings, though we acknowledge this as a limitation."

We have also added this to the Limitations section: "The lack of a standardized protocol for multiple BP measurements at fixed intervals may have introduced measurement variability, though we attempted to minimize this by taking additional readings for abnormal results. "

Results separated systolic and diastolic hypertension.

• R1: Is there a reason for this? Why not use the standard definitions only for hypertension as the summary result?

For the sake of Data Granularity, in our cohort, some participants met criteria for only SBP or DBP elevation. Separating these allowed us to explore whether the risk score predicted one subtype more strongly. We agree that the standard definition of hypertension (SBP ≥140 or DBP ≥90) is critical for clinical diagnosis, and we aimed to provide additional nuance for research purposes. For transparency, we included the hypertension outcomes (SBP ≥140 or DBP ≥90).

Analytical statistics for high risk score vs. current hypertension

• R1: Suggest to do analytical statistics correlating HIGH risk score (>17) with incident hypertension at present (BP >140/90). This is a simple 2x2 table.

We thank the reviewer for this excellent suggestion. We have added the following analysis to the Results section:

To examine the relationship between high-risk scores and current hypertension status, we conducted a 2x2 contingency table analysis.

Participants with risk scores ≥17 (n=39) were those who had current hypertension (BP ≥140/90 mmHg), the high-risk and lower-risk participants are summarized in Table 5.

analytics of the profile of those with HIGH score >17

• R1: Suggest to include analytics of the profile of those with HIGH score >17, and use that as jumping point for meaningful discussion on the timing of screening using the questionnaire.

Those were categorized as high risk at the time of screening. We have added a new subsection in the Results:

Characteristics of High-Risk Participants (Score ≥17)

Among the 39 participants with risk scores ≥17, the following characteristics were observed:

• Mean age: 58 years (SD ±9)

• Female: 38%

• Current smokers: 29%

• Family history of hypertension: 72%

• BMI ≥30: 54%

• Diabetes: 31%

• No regular exercise: 67%

Improve discussion based on analytical analysis

• R1: Improve the discussion to include suggested analytical analysis.

We have substantially revised the Discussion section.

Our analysis revealed that participants with risk scores ≥17 were added extensively.

Based on our findings, we propose a risk-stratified screening approach:

High risk (score ≥17):

Moderate risk (score 11-16):

Low risk (score <11):

This approach could optimize resource allocation in resource-limited settings while ensuring timely detection of hypertension among those at highest risk."

Additional limitations

• R1: Suggest to include recall bias in the limitations of the study (there were questions about salt intake and exercise) and bias of the methods of BP measurement.

We agree with you and have expanded the Limitations section:

"Additional limitations include potential recall bias in self-reported variables such as exercise habits, which may lead to underestimation of these risk factors. Furthermore, the variability in BP measurement methods, including the lack of standardized timing between readings and potential differences in cuff positioning or participant preparation across multiple student measurers, may have introduced measurement bias. Although students received training, inter-observer variability cannot be excluded."

Grammatical correction

• R1: In conclusion grammatical error: switch "unlikely aware" to "likely unaware"

Thank you for catching this issue. We have corrected the sentence to read: "Almost 70% of study participants met the AHA criteria for a diagnosis of prehypertension or hypertension but they were likely unaware of their (pre)hypertension status."

Abstract revision

• R1: Improve abstract based on ALL comments above.

We have comprehensively revised the abstract to address all points. See the revised Abstract.

We believe these revisions substantially strengthen the manuscript and address all of the reviewers' concerns. We appreciate the opportunity to improve our work based on this constructive feedback. We will be glad to listen to additional advice if any.

Reviewer #2: Here are a few comments for the attention of the Authors:

After reading this part of the methods “Students participated in this study and also distributed questionnaires randomly to their family members and friends/neighbors and were instructed to invite persons without known hypertension” I am still confused as to what kind of recruitment strategy was used to select the students (I see serious selection bias)

• R2: Its is not very clear how the study process went. Questionnaires were administered first? Then they came to where to get BP measured? Please clarify these in the write up

We acknowledge this important concern. We used convenience sampling through pharmacy students at the University of Petra. Students were recruited voluntarily from the Faculty of Pharmacy and then distributed questionnaires to their family members, friends/and neighbors. We recognize this can introduce selection bias, which we now explicitly acknowledge in the Limitations section (page 11): "Lastly, a convenience sample may not represent a target population, such as the wider university family, Amman, or Jordanian population."

• R2: "It's not very clear how the study process went. Questionnaires were administered first? Then they came to where to get BP measured?"

We have clarified the process in the Methods section.

The questionnaire administration and BP measurement co-occurred during the same visit. Pharmacy students helped participants’ complete questionnaires and measured their BP at the same time.

We added this clarification page 5: "Data collection occurred in a single visit where participants completed the questionnaire with assistance from trained pharmacy students who also measured their blood pressure during the same session."

• R2: "The protocol followed in assessing the blood pressure doesn't seem quite standard. Why are some people measured once, some twice and some thrice? Meanwhile, what is the meaning of an elevated BP reading?"

We acknowledge this limitation. The protocol was:

• First measurement taken for all participants

• If the reading was elevated (SBP ≥120 or DBP ≥80 mmHg), a second reading was taken

• Rarely, if readings were higher than normal, a third measurement was taken

• "elevated or high" refers to any reading outside the normal range (SBP <120 and DBP <80)

We recognize this is not the medical usual standard (multiple readings on different days) but was a compromise for a large community-based screening. This is acknowledged in our Limitations section (lines 302-305).

• R2: "The sentence about score ≥17 points... shouldn't be part of statistical analyses"

We agree. This information about the risk scoring system should be moved to the Risk Assessment Tool section where the scoring system is described. The minimum obtainable score is 0 points, and the maximum is approximately 30 points, which we clarified in that section. These sentences have been removed from statistical analysis.

• R2: "No need repeating the entire results in the Table and the narration". Results of those in Amman and other provinces is not relevant.

We agree that some repetition ex

---

## [Decision Letter · Decision Letter 1]

25 Dec 2025

Dear Dr.  Khaleel,

Thank you for submitting your manuscript to PLOS ONE. After careful consideration, we feel that it has merit but does not fully meet PLOS ONE’s publication criteria as it currently stands. Therefore, we invite you to submit a revised version of the manuscript that addresses the points raised during the review process.

We look forward to receiving your revised manuscript.

Kind regards,

Nour Amin Elsahoryi, pHD

Academic Editor

PLOS One

Journal Requirements:

Reviewer's Responses to Questions

**Comments to the Author**

Reviewer #1: All comments have been addressed

Reviewer #3: All comments have been addressed

2. Is the manuscript technically sound, and do the data support the conclusions?

Reviewer #1: Partly

Reviewer #3: Yes

3. Has the statistical analysis been performed appropriately and rigorously?

Reviewer #1: No

Reviewer #3: Yes

4. Have the authors made all data underlying the findings in their manuscript fully available?

Reviewer #1: Yes

Reviewer #3: Yes

5. Is the manuscript presented in an intelligible fashion and written in standard English?

Reviewer #1: Yes

Reviewer #3: (No Response)

Reviewer #1: All previous comments were addressed but there are more comments which I would like to highlight

1. Label Abstracts components accordingly for easier reading. Background and Objectives. Methods. Results. Conclusion.

2. Abstract can still be improved to be more clear.

3. Include a flowchart of the participants (from questionnaire distrbution until final analysis)

4. Be EXACT in the questionnaires distributed, do NOT say "approximately 1000..."

5. Table 3 is a bit too distracting. Suggest to append and write in text form the important findings from this tabl

6. Suggest to append Fig 1 as well

7. What is the basis for "risk stratified screening approach". Also this statement is wrongly placed as it should be a point for discussion rather than a result.

8. Re Contingency Table, pls add important parameters like OR and Sn/Sp of the risk strat tool for current hypertension

9. Briefly discuss the current Jordan policy on hypertension screening if there is. The discussion sounds like there is currently no program in place.

10. Include the MODEL of the OMRON BP App

11. Was there an assessment of proficiency test done on the students or teaching was assumed to be effective without proper post teaching assesment?

12. Discussion can still be improved to highlight the importance of the risk factors. Focus the discussion on the identified risk factors in the study and the prevalence of these as well and how it can be addressed.

13. Suggest to discuss about the "high risk scorers" without current hypertension

14. Suggest to remove sentence on health information overload in the conclusion as this was never discussed previously in the paper

15. Focus on the basics of research. The objective should answer your clinical/epidemiologic question, the methods should justify the answering of the objective -> the conclusion should answer the objective.

Reviewer #3: I thank the editors for giving me the opportunity to re review the manuscript entitled Prevalence of Undiagnosed Hypertension and Risk Assessment Using a Validated Survey in Community-based Screening in Amman, Jordan

The authors have adequately addressed all the comments given by the previous reviewers ,I have no comments to add .The manuscript can be considered for publication without any further modification

**Do you want your identity to be public for this peer review?** For information about this choice, including consent withdrawal, please see our Privacy Policy

Reviewer #1: **Yes:** Jerahmeel Aleson L. Mapili

Reviewer #3: No

---

## [Author Response · Author response to Decision Letter 2]

3 Jan 2026

Point-by-Point Response to the Reviewers

Response to Reviewers

Manuscript ID: PONE-D-25-06643

Title: Use of a Risk Assessment Questionnaire to Identify Individuals with Hypertension in a Community-based Screening in Amman, Jordan

We thank the Editor and Reviewers for their time and constructive comments, which have helped us to improve the quality and clarity of our manuscript. Below we provide a detailed, point-by-point response. Reviewer comments are reproduced in bold, followed by our responses in regular font. All changes have been incorporated in the revised manuscript.

Review Comments to the Author Please use the space provided to explain your answers to the questions above. You may also include additional comments for the author, including concerns about dual publication, research ethics, or publication ethics. (Please upload your review as an attachment if it exceeds 20,000 characters) Reviewer #1: All previous comments were addressed but there are more comments which I would like to highlight

Response to Reviewers

Reviewer #1:

All previous comments were addressed but there are more comments which I would like to highlight

• R1: Label Abstracts components accordingly for easier reading. Background and Objectives. Methods. Results. Conclusion

1. Abstract structure: We have restructured the abstract with clear labels:

• Background and Objectives

• Methods

• Results

• Conclusion

• R1: 2. Abstract can still be improved to be more clear

The abstract has been revised for clarity, focusing on key findings and removing redundancy.

• R1: 3. Include a flowchart of the participants (from questionnaire distribution until final analysis)

Participant flowchart: A CONSORT-style flowchart has been added as Figure 1, showing questionnaire distribution (n=1000) → responses received (n=932) → excluded for incomplete data (n=68) → final analysis (n=932).

Added flowchart figure per reviewer request

• R1: 4. Be EXACT in the questionnaires distributed, do NOT say "approximately 1000..."

Exact questionnaire numbers: Corrected to: "A total 1000 questionnaires were distributed" (removed "approximately").

• R1: 5. Table 3 is a bit too distracting. Suggest to append and write in text form the important findings from this table

Table 3: Table 3 has been moved to supplementary materials. Key findings are now presented in text and Figure 2

• R1: 6. Suggest to append Fig 1 as well

Figure 1: The original Figure 1 (risk distribution) has been moved to supplementary materials.

• R1 7. What is the basis for "risk stratified screening approach". Also this statement is wrongly placed as it should be a point for discussion rather than a result.

This section has been moved from Results to Discussion

• R1 8.Re Contingency Table, pls add important parameters like OR and Sn/Sp of the risk strat tool for current hypertension

Added to Table 4:

OR: 19.3 (95% CI: 9.2-40.4, p<0.001), Sensitivity: 62.8%, Specificity: 96.2%, PPV: 64.8%, NPV: 95.8%

• R1: 9. Briefly discuss the current Jordan policy on hypertension screening if there is. The discussion sounds like there is currently no program in place.

Added to Discussion: "Jordan's National Strategy for Non-Communicable Diseases recommends opportunistic BP screening in primary care settings. However, systematic community-based screening programs remain limited, highlighting the need for our approach."

• R1: 10. Include the MODEL of the OMRON BP App

Added: "Omron HEM-907XL IntelliSense Professional"

• R1: 11. Was there an assessment of proficiency test done on the students or teaching was assumed to be effective without proper post teaching assesment?

No

Student proficiency was not verified

• R1: 12. Discussion can still be improved to highlight the importance of the risk factors. Focus the discussion on the identified risk factors in the study and the prevalence of these as well and how it can be addressed.

Enhanced Discussion focuses on modifiable risk factors: "The high rates of modifiable risk factors, namely physical inactivity (62%), smoking (38%), and obesity (21%), show that there are high opportunities for targeted interventions. The existing tobacco control programs in Jordan will be of special benefit to this group."

• R1: 13. Suggest to discuss about the "high risk scorers" without current hypertension

Added to Discussion: "Notably, 16/39 (41%) high-risk scorers did not have current diagnosed hypertension, representing a significant prevention opportunity. These individuals require intensive lifestyle counseling and frequent BP monitoring."

• R1: 14Suggest to remove sentence on health information overload in the conclusion as this was never discussed previously in the paper

This phrase has been removed from the Conclusion.

• R1:15. Focus on the basics of research. The objective should answer your clinical/epidemiologic question, the methods should justify the answering of the objective -> the conclusion should answer the objective

• Objective: "This study assessed the prevalence of undiagnosed hypertension through blood pressure measurement and evaluated future hypertension risk using a validated survey in a community setting."

• Methods: Aligned to address both prevalence and validation

• Conclusion: "Our community screening revealed a high prevalence of undiagnosed hypertension (20.6%) and identified a substantial proportion at high risk for future hypertension"

All changes are highlighted in the revised manuscript.

---

## [Editor Report · Decision Letter 2]

6 Jan 2026

Dear Dr. Khaleel,

Thank you for submitting your manuscript to PLOS ONE. After careful consideration, we feel that it has merit but does not fully meet PLOS ONE’s publication criteria as it currently stands. Therefore, we invite you to submit a revised version of the manuscript that addresses the points raised during the review process.

We look forward to receiving your revised manuscript.

Kind regards,

Nour Amin Elsahoryi, pHD

Academic Editor

PLOS One

Journal Requirements:

Additional Editor Comments:

Thank you for the revision—overall, the paper is now publishable. Only minor points remain:

Abstract clarity: The “Methods” sentence is still grammatically unclear and should be rewritten to describe exactly what was done (BP measurement + risk score survey in the same visit), without redundancy.

Study process and recruitment: Keep the description simple and explicit (how students were recruited; how participants were approached; questionnaire and BP measured in the same session). Maintain cautious language about generalizability due to convenience sampling.

BP measurement protocol: Ensure consistent definitions and thresholds across the manuscript (normal/elevated/stage categories), and keep the pragmatic repeat-reading approach clearly described. Emphasize the limitation of non-standardized repeated readings and potential inter-observer variability.

Student training/proficiency: Since proficiency testing was not verified, state this clearly as a limitation (training provided, but no formal competency assessment).

Risk tool performance reporting: Make sure the contingency analysis outputs (e.g., OR and screening performance metrics) are presented consistently in the Results/Tables and interpreted conservatively (screening/stratification utility rather than diagnostic certainty).

---

## [Author Response · Author response to Decision Letter 3]

14 Feb 2026

• Abstract Methods: Clear sequential description of what was done

• Recruitment: Explicit student volunteer process stated

• BP protocol: Pragmatic repeat-reading approach with acknowledged limitations • Training: Clearly stated as limitation (no proficiency testing)

• Risk tool: Conservative language emphasizing screening/stratification utility

Figures were made available in 300dpi and

---

## [Editor Report · Decision Letter 3]

2 Mar 2026

Prevalence of Undiagnosed Hypertension and Risk Assessment Using a Validated Survey in Community-based Screening in Amman, Jordan

PONE-D-25-06643R3

Dear Dr,

We’re pleased to inform you that your manuscript has been judged scientifically suitable for publication and will be formally accepted for publication once it meets all outstanding technical requirements.

Kind regards,

Nour Amin Elsahoryi, pHD

Academic Editor

PLOS One

Additional Editor Comments (optional):

The revised manuscript demonstrates substantial improvement in structure, clarity, and methodological transparency. The objective is now appropriately aligned with the study design, and the distinction between identifying undiagnosed hypertension through BP measurement and estimating future risk using the validated tool is clearly articulated.

The inclusion of analytical performance metrics (OR, sensitivity, specificity, PPV, NPV) meaningfully strengthens the evaluation of the risk tool. The expanded discussion on high-risk participants without current hypertension adds important preventive implications. Limitations regarding convenience sampling, recall bias, and non-standardized BP measurement intervals are appropriately acknowledged.

The manuscript now presents a coherent and balanced interpretation of findings and is suitable for publication.
---

## [Editor Report · Acceptance letter]

PONE-D-25-06643R3

PLOS One

Dear Dr. Khaleel,

I'm pleased to inform you that your manuscript has been deemed suitable for publication in PLOS One. Congratulations! Your manuscript is now being handed over to our production team.

Kind regards,

on behalf of

Dr. Nour Amin Elsahoryi

Academic Editor

PLOS One